# Decorin Concentrations in Aqueous Humor of Patients with Diabetic Retinopathy

**DOI:** 10.3390/life11121421

**Published:** 2021-12-17

**Authors:** Shermaine W. Y. Low, Tanuja Vaidya, Santosh G. K. Gadde, Thirumalesh B. Mochi, Devesh Kumar, Iris S. Kassem, Deborah M. Costakos, Baseer Ahmad, Swaminathan Sethu, Arkasubhra Ghosh, Shyam S. Chaurasia

**Affiliations:** 1Ocular Immunology and Angiogenesis Lab, Department of Ophthalmology and Visual Sciences, Medical College of Wisconsin, Milwaukee, WI 53226, USA; wlow@mcw.edu (S.W.Y.L.); dkumar@mcw.edu (D.K.); ikassem@mcw.edu (I.S.K.); dcostakos@mcw.edu (D.M.C.); bahmad@mcw.edu (B.A.); 2GROW Research Laboratory, Narayana Nethralaya, Bangalore 560099, India; tanuja.vaidya@narayananethralaya.com (T.V.); swaminathansethu@narayananethralaya.com (S.S.); 3Vitreo-Retina Services, Narayana Nethralaya, Bangalore 560010, India; drsantoshgk@narayananethralaya.com (S.G.K.G.); thirumaleshmb@narayananethralaya.com (T.B.M.); 4Pediatric Ophthalmology, Eye Institute, Medical College of Wisconsin, Milwaukee, WI 53226, USA; 5Vitreoretinal Surgery, Eye Institute, Medical College of Wisconsin, Milwaukee, WI 53226, USA

**Keywords:** diabetic retinopathy, decorin, SLRP, aqueous humor, angiogenesis, inflammation, visual acuity

## Abstract

Diabetic retinopathy (DR) is a microvascular complication of diabetes in the retina. Chronic hyperglycemia damages retinal microvasculature embedded into the extracellular matrix (ECM), causing fluid leakage and ischemic retinal neovascularization. Current treatment strategies include intravitreal anti-vascular endothelial growth factor (VEGF) or steroidal injections, laser photocoagulation, or vitrectomy in severe cases. However, treatment may require multiple modalities or repeat treatments due to variable response. Though DR management has achieved great success, improved, long-lasting, and predictable treatments are needed, including new biomarkers and therapeutic approaches. Small-leucine rich proteoglycans, such as decorin, constitute an integral component of retinal endothelial ECM. Therefore, any damage to microvasculature can trigger its antifibrotic and antiangiogenic response against retinal vascular pathologies, including DR. We conducted a cross-sectional study to examine the association between aqueous humor (AH) decorin levels, if any, and severity of DR. A total of 82 subjects (26 control, 56 DR) were recruited. AH was collected and decorin concentrations were measured using an enzyme-linked immunosorbent assay (ELISA). Decorin was significantly increased in the AH of DR subjects compared to controls (*p* = 0.0034). AH decorin levels were increased in severe DR groups in ETDRS and Gloucestershire classifications. Decorin concentrations also displayed a significant association with visual acuity (LogMAR) measurements. In conclusion, aqueous humor decorin concentrations were found elevated in DR subjects, possibly due to a compensatory response to the retinal microvascular changes during hyperglycemia.

## 1. Introduction

Diabetic retinopathy (DR) is a sight-threatening microvascular complication of diabetes mellitus (DM), with a prevalence as high as 60% in patients with more than 10 years and 90–99% with 20 years of DM [1]. In 2020, the estimated number of adults suffering from DR worldwide was 103.12 million. These numbers are predicted to rise to 160.50 million by 2045 [2]. DR is a progressive disease divided into nonproliferative DR (NPDR) and proliferative DR (PDR). In NPDR, patients are in the early stages of the disease and display signs of increased vascular permeability, microaneurysms, hemorrhages, hard exudates, and sometimes macular edema. In the more advanced stage of DR, known as PDR, patients can also have severe vision impairment due to aberrant neovascularization with blood vessel episodic rupture that can cause vitreous hemorrhage and eventual retinal detachment. Tractional forces due to fibrosis may also result in retinal detachment. DR prediction tools rely on traditional DM risk factors: genetic factors, duration of diabetes, hyperglycemia, hypertension, and hemoglobin-A1c (HbA1c) concentrations [1,3,4]. These factors, however, remain insufficient as predictors for sight-threatening DR. Control of blood glucose, glycosylated hemoglobin, and blood pressure for preventing DR progression remains controversial, with several studies suggesting that optimal control reduces DR progression [5,6], whilst others report opposing findings [7,8]. Further analysis of the biochemical changes in human eyes during DR progression may uncover new predictors for the disease.

Current treatment strategies for DR focus on the later stages of the disease. Intravitreal injections of antiangiogenic and anti-inflammatory drugs are given to patients with DR. However, these drugs have high treatment burden [9] and pose a risk of endophthalmitis. Furthermore, anti-vascular endothelial growth factor (VEGF) agents and intravitreal steroids usually require multiple injections, limiting their use due to financial burden and poor patient compliance. Traditional laser photocoagulation is used for the treatment of PDR and diabetic macular edema (DME). However, the procedure permanently damages the treated retina and results in loss of visual field [10]. The absence of early-stage DR treatment strategies presents a challenge for the management of disease progression. Moreover, recurrent DR remains rampant, with some patients responding poorly to current treatment options despite repeated intravitreal injections. A deeper understanding of the pathogenesis of DR is necessary to develop novel therapeutic options for the optimization of current treatment strategies.

Decorin may be a potential therapeutic agent as it is reported to be significantly upregulated during obesity and type 2 diabetes [11]. In addition, previous studies have described its protective role in other diabetes-related comorbidities, including diabetic cardiomyopathy [12] and nephropathy [13]. In chronic hyperglycemia, inflammatory and angiogenic factors play a crucial role in the advancement of DR pathophysiology. Decorin is a class I small leucine-rich proteoglycan (SLRP) that acts as a pan-receptor tyrosine kinase inhibitor [14,15,16] and regulates inflammatory, angiogenic, and fibrotic pathways. In the eye, decorin prevents retinal pigmented epithelium (RPE) barrier disruption by reducing p38 MAPK phosphorylation, decreasing apical-basolateral permeability, and upregulating the expression of tight junction proteins, such as occludin and zonula occludens-1 [17]. Additionally, decorin can play a role in fibrillogenesis by inhibiting epithelial–mesenchymal transition (EMT) and fibrosis in spontaneously arising retinal pigment epithelium (ARPE-19) cells using an in vitro proliferative vitreoretinopathy (PVR) model [18]. ARPE-19 cells treated with high concentrations of decorin reduced expression of basement-membrane-associated proteins, such as fibronectin, laminin, vimentin, collagen I, and α-smooth muscle actin. Moreover, intravitreal injection of decorin significantly reduces fibrosis in rabbits with experimentally induced PVR [19], suggesting its potential role in modulating the extracellular matrix to prevent DR-induced tractional retinal detachment. We hypothesize that decorin plays a regulatory role in modulating ocular inflammatory and angiogenic responses during DR progression. Therefore, this study aims to examine the correlation between aqueous humor decorin concentrations to the clinical severity of DR.

## 2. Materials and Methods

All protocols were conducted in accordance with the Declaration of Helsinki and approved by the Narayana Nethralaya Institutional Ethics Board, as per the guidelines of Indian Council of Medical Research (ICMR), New Delhi, India. Subjects were recruited for the study post informed written consent (EC no. C/2017/11/03).

### 2.1. Study Subjects

This cross-sectional study included subjects attending the Narayana Nethralaya Hospital, Bangalore, India, for cataract surgery. Detailed medical history was collected by trained medical professionals, including age and sex. Inclusion criteria for controls were cataract patients who had come to Narayana Nethralaya for cataract surgery with no retinal disorders. Exclusion criteria for control included: (i) subjects with diabetes mellitus, (ii) coexisting fibrovascular conditions, such as proliferative diabetic retinopathy (PDR), age-related macular degeneration (AMD), epiretinal membrane (ERM), or retinal vessel occlusions (RVO), (iii) intraocular and inflammatory conditions, such as endophthalmitis or uveitis, and iv) degenerative conditions, such as glaucoma, or inherited diseases, such as retinitis pigmentosa or leber congenital amaurosis (LCA). Two control subjects had existing systemic hypertension.

Inclusion criteria for DR subjects included type 2 diabetic patients with clinically diagnosed DR which was confirmed with fundus imaging (TRC 50DX, Topcon, Japan). Exclusion criteria for DR patients included: (i) subjects with retinal edema caused by other retinal conditions, such as AMD, RVO, and other intraocular and inflammatory diseases, (ii) coexisting vitreoretinal interface issues, such as vitreomacular traction (VMT), ERM, and coexisting lamellar macular holes (MH), (iii) subjects who underwent vitrectomy or recent intraocular surgeries or procedures (<3 months), such as cataract or YAG capsulotomy, and (iv) degenerative conditions, such as glaucoma, and inherited diseases, such as retinitis pigmentosa, LCA, Stargardt disease, and cone–rod dystrophy. Twenty-six DR subjects had existing systemic hypertension, six had kidney problems, and four had heart problems.

### 2.2. Ocular Examinations

Ocular examinations, including slit lamp biomicroscopy, LogMAR chart for visual acuity, fundus photography, and optical coherence tomography, were performed on all subjects prior to aqueous humor extraction and cataract surgery.

### 2.3. Classification of Diabetic Retinopathy

DR grading was performed according to the Early Treatment Diabetic Retinopathy Study (ETDRS) classification system (Table 1A,B) [20,21] and the Gloucestershire Diabetic Eye Screening Program (Table 2).

### 2.4. Response to Treatment

Subjects were categorized based on the history of medication and disease resolution following ongoing therapy. Subjects with no prior treatment were grouped as treatment naïve (TN). Subjects who responded well to anti-VEGF/steroid treatment with respect to reduction in central sub-field thickness (CST) ≥ 100 µm from the baseline after 1 intravitreal injection and CST < 320 μm were grouped as treatment responders (TRes). Subjects that failed to achieve a CST of <320 μm and/or 10% reduction in CST after anti-VEGF/steroid treatment were grouped as treatment nonresponders (TnR). A recurring macular edema of >320 μm after a total resolution and injection-free period of ≥3 months and subjects who had further complications despite receiving therapy were categorized as treatment recurrent (TRec).

### 2.5. Aqueous Humor Collection

Aqueous humor was collected by anterior chamber paracentesis with a syringe with a 30-gauge needle to access the anterior chamber through the peripheral cornea to aspirate 50–100 μL of aqueous humor. Sample collection was performed before cataract surgery by the surgeons. All the samples were stored at −80 °C for a maximum of 11 months before further processing.

### 2.6. Decorin Measurements

Decorin concentrations were measured using Decorin Duoset ELISA kit from R&D Systems (Minneapolis, MN, USA) following manufacturer’s instructions. Aqueous humor samples were diluted in 1:5 ratio with 4 °C 1X PBS. In brief, capture decorin antibody was coated to a 96-well microplate overnight. The next day (12–15 h), blocking agent (reagent diluent) was added for 1 h at room temperature, and then samples/standards were added for 2 h at room temperature. After incubation, the microplate was washed with wash buffer, and detection antibody was added for 2 h, followed by streptavidin–HRP for an additional 20 min at room temperature. The optical density at 450 nm with wavelength correction at 540 nm was measured after the addition of substrate solution for 20 min in the dark and the addition of stop solution. Aqueous humor concentrations were described in pg/mL.

### 2.7. Statistical Analysis

Decorin concentrations have been transformed to log scale using GraphPad Prism 9.0 software (GraphPad Software, San Diego, CA, USA). Patient demographics were compared using Wilconxon rank-sum test and Chi-square test. Decorin levels between control and DR groups were compared using linear regression, Student’s *t*-test, and Kruskal–Wallis one-way analysis of variants with Tukey’s test using SAS 9.4 software (SAS Institute, Cary, NC, USA). All the data points, including outliers, were analyzed and violin plots were constructed using GraphPad to provide better distribution and density, as well as to plot the center, spread, and any outlier present in the data set. *p* < 0.05 was statistically significant.

## 3. Results

### 3.1. Study Cohort Subject Demographics

In this study, a total of 26 control subjects and 56 DR subjects were recruited based on the inclusion criteria mentioned above. There was no significant difference between the groups in terms of demographics. Control subjects were 40 to 80 years of age (median = 61), and DR subjects were 42 to 89 years of age (median = 62). Sex distribution was 18/8 (M/F) in control subjects and 43/13 (M/F) in DR subjects (Table 3). An overall summary of sample numbers has been included in Appendix A.

### 3.2. Decorin Concentrations Are Increased in Subjects with Diabetic Retinopathy

Average decorin concentrations in the DR group were significantly high compared to the control group (Figure 1A; control, 3.592 pg/mL; DR, 3.840 pg/mL; *p* = 0.0034). Subjects were then divided based on sex, and decorin concentrations between control and DR subjects were analyzed (Figure 1B–D, Appendix A). No significant difference was observed between female DR and control subjects (Figure 1B). Male DR subjects showed significantly increased concentrations of decorin compared to controls (Figure 1B; *p* = 0.0001). No significant difference was observed between female and male control subjects (Figure 1C). Male DR subjects had significantly higher decorin concentrations compared to female DR subjects (Figure 1D; *p* = 0.0003). Further analysis will be required with a larger sample size to determine if these changes are sex-driven.

The study cohort was divided into different DR groups based on ETDRS and Gloucestershire classifications. When DR subjects were separated into their ETDRS scores, no significant difference was observed between class 1–4. Post hoc Tukey’s analysis displayed a significant difference between controls and Class 4 subjects in ETDRS classification (Figure 2A, *p* = 0.0009). When divided based on the Gloucestershire classification system, no significant difference in decorin concentration between groups R1–R3 was observed. A significant difference was observed between the control and group R3M0 after Tukey’s analysis (Figure 2B, *p* = 0.0061). Tukey’s pairwise comparison data for ETDRS and Gloucestershire classifications can be found in Appendix A, respectively.

### 3.3. Decorin Concentrations Are Increased in Diabetic Retinopathy Subjects Regardless of Their Response to Treatment

We further classified the subjects into different groups based on their response to DR treatment (Figure 3). DR subjects were split into four groups: treatment naïve (TN), treatment responders (TRes), treatment recurrent (TRec), and treatment nonresponders (TnR). No significant difference was observed between the four different DR treatment groups (*p* > 0.05; TN, TRes, Trec, TnR). Decorin concentrations between nonresponders and the control group were significantly different under Tukey’s post hoc analysis (*p* = 0.0038, Appendix A).

### 3.4. Decorin Concentrations Are Positively Correlated with Deteriorating Visual Acuity

Next, correlation between decorin concentration and visual acuity was examined using a simple linear regression with patient logarithm of the minimum angle of resolution (LogMAR) measurements. The scatterplots show that decorin concentration and LogMAR have a statistically significant linear relationship that is positively correlated (Figure 4; Overall r = 0.24; *p* = 0.04).

Correlation between decorin concentration and LogMAR measurements were then divided based on the different sexes. Appendix A shows that the positive correlation between decorin concentrations and visual acuity in the overall correlation plot (Figure 4) are mainly driven by the male population. However, it is difficult to conclude if the changes in decorin concentrations are sex-driven due to the small sample size for the female population compared to males.

## 4. Discussion

This study demonstrates that the decorin concentrations in aqueous humor of DR patients were significantly elevated compared to the control subjects without DR. The presence of abnormal microvasculature in DR may be associated with increased concentrations of intraocular decorin (Figure 1A). When divided based on sex, the male DR population exhibited increased concentrations of decorin compared to controls (Figure 1B). Male DR decorin concentrations were also significantly higher than female DR decorin concentrations (Figure 1D). However, it is difficult to determine if these changes are sex-driven due to the small sample size of the female population. After post hoc analysis using Tukey’s adjustments, decorin concentrations were significantly increased in the more severe groups of DR classifications (Figure 2). Furthermore, late stages of DR correspond with worsening visual acuity due to retinal damage, floaters, and hemorrhage, where the association between aqueous humor decorin concentrations and patient visual acuity was observed. These results suggest that decorin concentrations were associated with the increase in DR severity. There was a correlation observed between visual acuity and decorin concentrations in control subjects. However, we can only speculate that this may be due to differences in cataract severity based on the limited data. Considering the proximity of the aqueous humor to the crystalline lens, cataract severity may be associated with changes in decorin concentrations. It will be interesting to study the relationship between decorin concentrations and severity of cataracts in future studies.

Decorin is known to play a critical role in regulating microvasculature during development and degenerative diseases in the retina [22,23,24]. Previous studies have shown that decorin can modulate VEGF and hypoxia-induced factor-1 alpha (HIF-1α) expression by blocking the Met pathway to prevent angiogenesis in retinal cells [25]. In addition, decorin was reported as an antagonistic ligand for VEGF receptor 2 [26], the primary receptor involved in vascular differentiation, capillary-like tube formation, and angiogenesis in vascular endothelial cells [27,28,29]. Further, decorin treatment to ARPE-19 cells cultured in high glucose media could reverse RPE barrier disruptions and increase tight junction proteins, such as occludin and zonula occludens-1 [17]. In the anterior segment of the eye, decorin gene therapy to rabbit corneal tissues restricted corneal neovascularization and reduced the expression of proangiogenic VEGF and monocyte chemoattractant protein-1 (MCP-1) transcripts [30]. These studies suggest that decorin reduces cell permeability and microvascular leakage in the ocular tissues. Decorin concentrations have also been elevated in ischemic retinas with damaged inner retinal layers and in PVR [18,22,31], suggesting that decorin levels are elevated during retinal disease. The dynamic interaction and balance of decorin in the eye may, thus, be crucial in maintaining physiological retinal integrity and microvasculature, and the increase in decorin levels in the aqueous humor may be associated with DR pathology.

Decorin has been reported to exhibit antiangiogenetic properties [30,32,33,34]. Chronic inflammation and neovascularization increase with the severity of DR and play a vital role in DR pathogenesis. Growth factors and inflammatory and angiogenic cytokines, such as IL-1β, IL-6, and VEGF in aqueous humor [35,36], transforming growth factor-beta (TGF-β) in aqueous humor [37] and serum [38], and tumor necrosis factor-alpha (TNF-α) in aqueous humor [39,40] and vitreous [41], have been significantly upregulated in diabetes and are associated with DR. TNFα induces disruption of the blood–retinal barrier and promotes endothelial cell permeability. TGF-β modulates the extracellular matrix by activating fibroblasts and preventing the synthesis of proteases, which help to degrade matrix components that are being overproduced [42]. Upregulation of these proinflammatory cytokines can result in vascular leakage commonly seen in PDR, and retinal fibrosis, which generates tractional forces on the delicate retina [43]. These tractional forces can ultimately lead to retinal detachment. Interestingly, intravitreal injection of decorin in a choroidal neovascularization (CNV) mouse model was able to suppress TGF-β, VEGF, and TNFα upregulation [44]. Decorin delivered intraperitoneally has also been reported to be able to reduce VEGF and TNFα immunoreactivity and the number of neovascular cell nuclei in an oxygen-induced retinopathy rat model [45]. In the present study, decorin concentrations were found to be significantly higher in the DR treatment nonresponder group after Tukey’s correction (Figure 3), suggesting that increased decorin concentrations are associated with DR pathology and could be a potential predictor for DR severity. Considering that decorin has been known to be an effective antagonist of TGF-β [46,47,48], and is able to reduce VEGF and TNFα expression in the retina, the elevated concentrations of decorin observed in the aqueous humor of DR patients could be a physiological compensatory response to combat inflammation and angiogenesis observed during DR progression.

Aqueous humor has been reported to be a good medium to study biochemical changes in the eye and has been directly correlated with vitreous humor biomarkers [49,50]. It is commonly used to determine intraocular changes associated with retinal pathology [51]. Decorin concentrations in the vitreous have been elevated in rhagmatogenous retinal detachment and PVR [18,31], suggesting that changes in aqueous decorin concetrations may be useful as a potential predictor for alterations that occur in the retina. Other studies have found vitreous concentrations of decorin to decline with age, irrespective of glaucoma or ocular hypertension in the eye [52]. This decline was suggested to be linked to increasing concentrations of active TGF-β2 during aging. However, diabetes is a multifactorial disease that influences multiple biochemical pathways. Further analysis will be needed to unveil the mechanistic and functional roles of decorin in various retinal diseases.

A limitation of this study is the sample size needed for the classification of DR groups. Future studies involving a larger sample size will be beneficial to understand the changes in decorin levels between different severities of DR. The study cohort was also unevenly divided in sex, limiting interpretation of the results. Additionally, routine collection of vitreous is highly contraindicated due to its invasiveness and access to vitreous sample is only possible as part of vitrectomy surgery. Therefore, decorin concentrations were only analyzed in aqueous humor samples. However, since aqueous humor has been reported to be directly correlated with vitreous humor biomarkers, collection and sampling of aqueous humor will be a more useful and feasible method for analysis of biochemical changes occurring in retinal diseases. Decorin’s expression and function in the retina have yet to be fully characterized. Previous studies have found decorin to be expressed in bovine aortic endothelial cells and human granulomatous tissue endothelial cells [53,54,55]. Further analysis to determine decorin-expressing retinal cells is required. Nonetheless, this study is the first to present associations between patient aqueous humor decorin concentrations and varying severities of DR classifications. The elevated concentrations of decorin in DR subjects suggest that decorin may play a role in modulating retinal extracellular matrix and microvascular integrity. Future studies are needed to characterize decorin’s functional roles in the maintenance of retinal microvasculature.

## 5. Conclusions

DR is a sight-threatening disease, which involves pathological modulation of the extracellular matrix and microvasculature, leading to multiple inflammatory and angiogenic responses during DR progression. This study reveals that decorin levels are elevated in subjects with DR, possibly due to a compensatory response to protect the retina against pathological remodeling in the extracellular matrix and retinal microvasculature. Our findings provide evidence for the importance of decorin in the retina. Further studies looking into decorin’s mechanisms of action will allow us to better understand its functional role in regulating retinal vasculature.

## Figures and Tables

**Figure 1 life-11-01421-f001:**
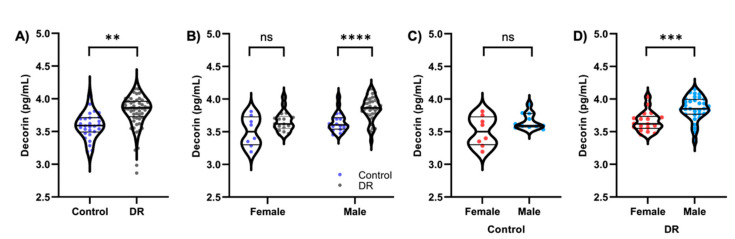
Comparison of decorin concentrations between control (*n* = 26) and DR (*n* = 56) subjects. Mean decorin concentration was significantly increased in subjects with DR (**A**). When divided based on sex, decorin concentrations were significantly increased in male DR subjects (**B**). No significant difference in decorin concentration was observed between female and male control subjects (**C**). Decorin concentrations in male DR subjects were significantly higher than in female DR subjects (**D**). Decorin concentrations have been transformed to log scale and plotted on a linear axis. Violin plots were constructed for the optimal data distribution, density, and inclusion of outliers in the data sets. ** *p* = 0.0034; *** *p* = 0.0003; **** *p* = 0.0001; ns = not significant.

**Figure 2 life-11-01421-f002:**
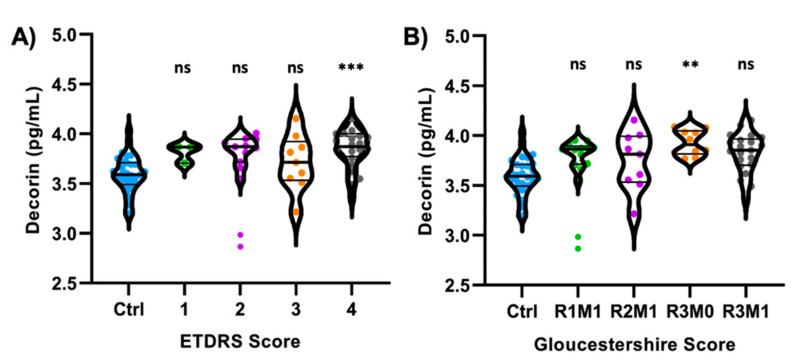
Comparison of decorin concentrations between control and DR subjects after dividing them into their ETDRS and Gloucestershire classifications. Decorin concentrations were significantly higher following post hoc Tukey’s analysis in ETDRS Class 4 (**A**) and Gloucestershire R3M0 (**B**) groups compared to controls. Decorin concentrations have been transformed to log scale and plotted on a linear axis. Violin plots were constructed for the optimal data distribution, density, and inclusion of outliers in the data sets. ** *p* = 0.0061; *** *p* = 0.0009; ns = not significant. ETDRS Ctrl (*n* = 26), Class 1 (*n* = 3), Class 2 (*n* = 13), Class 3 (*n* = 9), Class 4 (*n* = 31). Gloucestershire R0M0 (*n* = 26), R1M1 (*n* = 15), R2M1 (*n* = 9), R3M0 (*n* = 10), R3M1 (*n* = 22).

**Figure 3 life-11-01421-f003:**
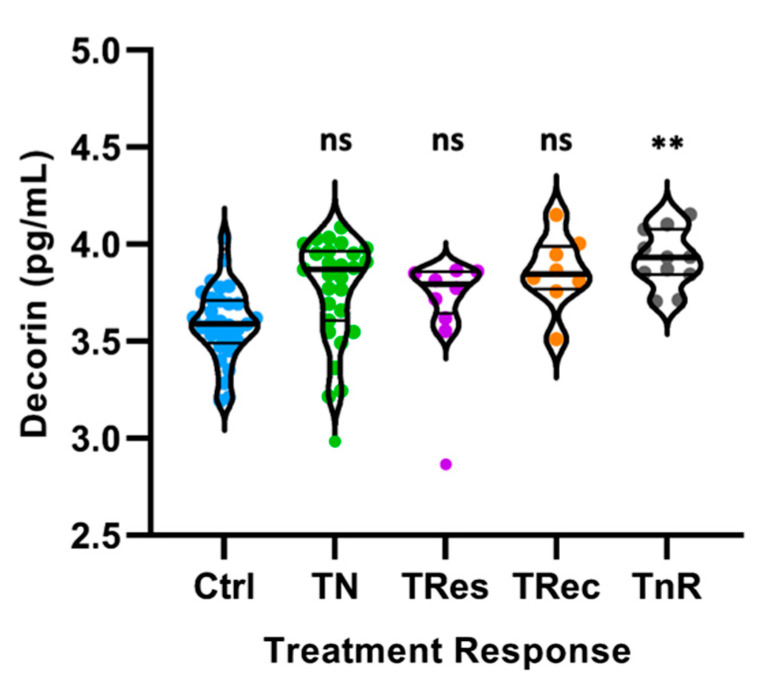
Comparison of decorin levels between controls (*n* = 26) and four DR treatment groups: treatment naïve (TN; *n* = 28), treatment responders (TRes; *n* = 9), treatment recurrent (TRec; *n* = 8), and treatment nonresponders (TnR; *n* = 11). Decorin concentrations have been transformed to log scale and plotted on a linear axis. Violin plots were constructed for the optimal data distribution, density, and inclusion of outliers in the data sets. Decorin concentrations were significantly elevated in the TnR group compared to controls after post hoc Tukey’s analysis. ** *p* = 0.0038; ns = not significant.

**Figure 4 life-11-01421-f004:**
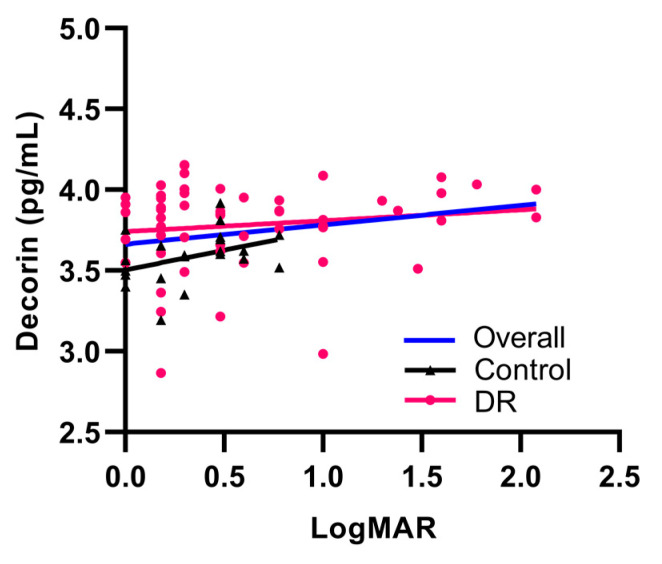
Correlation plot between decorin concentrations and patient visual acuity measured in LogMAR. Controls (*n* = 21) are represented by black triangles whilst DR subjects (*n* = 54) are represented by pink circles in the scatterplot. The blue line represents the overall linear relationship for all subjects (r = 0.24). The magnitude of overall association is mild (0.3 < |r| < 0.5). The black line represents the linear relationship for the control group (r = 0.38). The pink line represents the linear relationship for the DR group (r = 0.14).

**Table 1 life-11-01421-t001:** **A.** ETDRS classification. **B.** ETDRS definition of clinically significant macular edema (CSME).

A.		
ETDRS Class	DR Severity	Observable Findings
Control	No DR	No abnormalities
1	Mild NPDR	At least 1 microaneurysm or hemorrhage
2	Moderate NPDR	Microaneurysm or hemorrhage; or Venous beading; or Cotton wool spots; or Intraretinal microvascular abnormalities (IRMA)
3	Severe NPDR	Microaneurysm and hemorrhage in 4 quadrants; or Venous beading in 2 quadrants; or Severe IRMA in 1 quadrant
4	PDR	New vessel formation; or Vitreous hemorrhage; or Tractional retinal detachment
**B.**		
**CSME Class**	**Observable Findings**	
No CSME	No abnormalities	
CSME	Retinal thickening within 500 µm of the fovea Hard exudates within 500 µm of the fovea if associated with thickening of adjacent retina One or more zones of retinal thickening larger than 1500 µm that is within one disc diameter (1DD; 1500 µm) of the fovea	

**Table 2 life-11-01421-t002:** Gloucestershire classification.

Gloucestershire Class	Observable Findings
Control	No abnormalities
R1	Non-proliferative DR Microaneurysm(s) Hemorrhage Exudates
R2	Non-proliferative DR Multiple microaneurysms and hemorrhages Venous beading Intraretinal microvascular abnormalities
R3	Proliferative DR Vitreous hemorrhage New vessel formation Tractional retinal detachment
M0	No maculopathy
M1	Clinically significant macular edema, hemorrhage, or exudates within 1DD of the fovea

**Table 3 life-11-01421-t003:** Patient demographics in the study.

	Disease Category			
Variables	Total *N* = 82 (col %)	Control *N* = 26 (col %)	DR *N* = 56 (col %)	*p* Value
**Age** (years)				*0.881* ^W^
Median (min–max)	62.0 (40.0–89.0)	61.5 (40.0–80.0)	62.0 (42.0–89.0)	
Mean ± SD	62.5 ± 8.9	62.4 ± 9.1	62.5 ± 9.0	
**Sex**				*0.466* ^C^
Female	21 (25.6)	8 (30.8)	13 (23.2)	
Male	61 (74.4)	18 (69.2)	43 (76.8)	

^W^ Wilcoxon rank-sum test; ^C^ Chi-square test.

## Data Availability

Data are available upon request.

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
