# Peer review of "Decorin Concentrations in Aqueous Humor of Patients with Diabetic Retinopathy"

_life, 2021, doi:10.3390/life11121421_

Round 1

Reviewer 1 Report

Low and colleagues report increased decorin aqueous humor levels (ELISA) in patients suffering of diabetic retinopathy. This positive correlation was also associated with a lower visual acuity outcome, yet very mild. The work structured the analysis of the cohort of samples depending on multiple factors including severity of DR and response to treatment, showing significant differences in decorin levels only in high severity cases or in non-responders respectively. Authors suggest the relevance of decorin as a potential biomarker in DR.

This work is interesting given that decorin is an understudied protein regarding DR. Despite the work is presented as a (clinical) report, there is a very interesting potential towards more in-depth approaches and the molecular understanding of decorin in pathology. The paper is overall well written. There are, unfortunately, some problems regarding the experimental design of the study that need to be addressed. Some analyses are biased and would benefit to be complemented with additional variables given the relatively big cohort of patients (e.g. despite sex of the patients was recorded, there are no further analysis identifying sex-driven patterns; same applies for age).

Here are my main concerns and comments hoping the authors find them useful to improve their work:

  1. Authors should define DR consistently if reported in patients with T1D or T2D, as well as the years after disease was diagnosed if possible. This could be critical in the understanding of the complications and possibly correlated thus to decorin levels.

  1. In the exclusion criteria for control cases, there exist only criteria regarding eye pathology. In a study targeting DR this is unacceptable as systemic metabolic disease (fundamentally diabetes but also others such as obesity or family history) should be taken in consideration for control cases. This needs to be addressed retrospectively in the study if information is not available.

  1. Figure 1. Please use letters to define the distinct panels and thus be able to refer properly in both text and figure legend to the sections in the figure.

  1. Figure 1. Why there are two rows the same (R3M1) with different pictures? This needs to be clarified as the fundus and OCT images are clearly indicating very different phenotypes.

  1. Figure 1: Please add the amount of cases (N) per group in the cohort of the study to have the distribution of patients in a glance regarding the classification method. This is important since in Figure 3A-B you compare the distribution of the patients according to the classification method and results vary.

  1. Indicate if aqueous humor samples were collected by the same surgeon or not, and in which range of time (e.g. months, years). Long-periods of tissue bank storage should be reported.

  1. All references to gender (sociocultural) are misleading. As far as I understand, only sex (biological) is reported in this paper. Please follow sex and gender medicine guidelines to report this.

  1. Figure 2: Given the existence of information regarding sex in the cohort, authors should use this variable to analyze the data. It is true that cohort size limits statistical power when attending to sex, but for Figure 2 (full DR set) authors should display also graphs depicting sex-driven differences in decorin levels.

  1. Table 1: Despite it is implicitly clear, please indicate unit of age (years) in the table as well.

  1. Figure 3: I don’t understand why in the text is explained that no significant differences (except for Control vs. Class5 or Control vs. R3M1) and in the graph it is represented again the same statistical analysis done for Figure 2 (control vs. all DR groups together). And not only that, but when grouping all in the ETDRS scoring, 1 * is lost.

  1. Figure 4 does not represent the statistical analysis described in the text. Please refer the comparisons bar (condition) vs. bar (condition) without grouping.

  1. Figure 5. The authors hypothesize that decorin is associated with disorders such as obesity (introduction). Perhaps would be better to establish this correlation using only the DR samples or complementing the existing figure 5 with a “only DR” correlation (which looks even milder than when combined with control).

  1. Figure 5: Is there a sex-driven correlation of decorin with visual acuity? Is visual acuity itself having a sex-driven pattern?

  1. There exist reports of decorin levels reduced in aging and independently of pathology in many cases. Given that diabetes, and more specifically DR is in many cases age-related (T2D), could the authors explain or hypothesis why there is an increase beyond the compensatory role for angiogenesis? The global analysis of DR including non-angiogenic or low-inflammatory patients were also reported with higher decorin levels.

  1. There are some fundamental aspects of molecular biology needed in the context of disease and of this paper. Which cells produce decorin in the eye? And more specifically in the retina?

  1. Are there reports evidencing decorin in the vitreous? Given that DR is a retina disease, most of the by-products of the pathology lead to be highly present in the vitreous (direct contact). This is an important question for 2 main reasons: 1) there are not many reports correlating aqueous to vitreous proteomics and 2) if decorin constitutes a potential biomarker, access to aqueous humor is much more feasible than vitreous, and knowing if it is correlated would accelerate research and potentially diagnosis.

  1. Avoid repetition of results (stats) in discussion if already described in results.

  1. Decorin in eye disease is a very interesting paradigm. Authors report very well that decorin has been described to exert anti-angiogenic abilities. Following that line, it would be logical then to find decorin more present in NPDR or in patients that respond to therapy (thus having lesser inflammatory or angiogenic profile). Why is it in then that non-responders or severe PDR cases show higher levels? Despite the authors discuss this could be a compensatory mechanism, there are not hints or evidences for this counterintuitive explanation. Are there other works showing this experimentally? Please discuss more.

  1. Authors suggest in the discussion that decorin might modulate microvascular integrity, yet there is lack on data in this. Is there the possibility to extend this study to explore the imaging data of the patients and look for vascular features in the photographies?

Author Response

Reviewer 1

We really appreciate the comments and suggestions, which has improved the manuscript.

Below are the answers to reviewers comments:

  1. Authors should define DR consistently if reported in patients with T1D or T2D, as well as the years after the disease was diagnosed if possible. This could be critical in the understanding of the complications and possibly correlated thus to decorin levels.

We apologize for the mistake. The study was performed in type 2 diabetic (T2D) subjects. We have accordingly updated the manuscript. Mentioned on line 133, page 3.

  1. In the exclusion criteria for control cases, there exist only criteria regarding eye pathology. In a study targeting DR this is unacceptable as systemic metabolic disease (fundamentally diabetes but also others such as obesity or family history) should be taken in consideration for control cases. This needs to be addressed retrospectively in the study if information is not available.

We agree with the reviewer. The family history and other systemic disease history was taken into consideration as a routine for all the cases and especially during preoperative evaluation for the subjects. Exclusion criteria for control subjects have been edited to reflect any systemic metabolic diseases as mentioned on line 129, page 3.

  1. Figure 1. Please use letters to define the distinct panels and thus be able to refer properly in both text and figure legend to the sections in the figure.

Thanks for the suggestion. Figure 1 has been updated to include letters for the specific classifications. Modifications in text have been made in section 2.3.

  1. Figure 1. Why there are two rows the same (R3M1) with different pictures? This needs to be clarified as the fundus and OCT images are clearly indicating very different phenotypes.

Figure 1 has been updated as per the suggestion. Both the image sets belong to R3M1- showing macular edema in different conditions. The images are different as these belong to 2 different patients but both classified as R3M1. The second set has vitreous haemorrhage in addition to DME. Hence, we have removed one of the picture to avoid the confusion.

  1. Figure 1: Please add the amount of cases (N) per group in the cohort of the study to have the distribution of patients in a glance regarding the classification method. This is important since in Figure 3A-B you compare the distribution of the patients according to the classification method and results vary.

As suggested, cases (n) per group have been included into the figure 1 and 3 legends.

  1. Indicate if aqueous humor samples were collected by the same surgeon or not, and in which range of time (e.g. months, years). Long-periods of tissue bank storage should be reported.

The aqueous humor collection was performed by two surgeons (SGKG and TBM) on their OR days. The samples were collected over a period of one year and were stored for a maximum of 11 months before being processed.

  1. All references to gender (sociocultural) are misleading. As far as I understand, only sex (biological) is reported in this paper. Please follow sex and gender medicine guidelines to report this.

All references to gender has been updated throughout the text to reflect biological sex instead.

  1. Figure 2: Given the existence of information regarding sex in the cohort, authors should use this variable to analyze the data. It is true that cohort size limits statistical power when attending to sex, but for Figure 2 (full DR set) authors should display also graphs depicting sex-driven differences in decorin levels.

We agree with the reviewer that the cohort size limits our study and provide enough statistical power to analyze data based on sex.  As per suggestion we have included a supplementary table A10.

  1. Table 1: Despite it is implicitly clear, please indicate unit of age (years) in the table as well.

We have included age (years) in the table.

  1. Figure 3: I don’t understand why in the text is explained that no significant differences (except for Control vs. Class5 or Control vs. R3M1) and in the graph it is represented again the same statistical analysis done for Figure 2 (control vs. all DR groups together). And not only that, but when grouping all in the ETDRS scoring, 1 * is lost.

Thanks for suggestion. We have changed the graph to represent correctly what is said in the text. ANOVA suggested that there was difference between the groups and post hoc analysis performed with Tukey’s test suggested that Decorin concentration in ETDRS class 5 and Gloucestershire R3M0 was significantly higher than their respective control groups (Supplementary Table A1 and A2).

  1. Figure 4 does not represent the statistical analysis described in the text. Please refer the comparisons bar (condition) vs. bar (condition) without grouping.

We have modified the text. All DR subjects fall under the four treatment groups: Treatment naïve (TN), Treatment responders (TRes), Treatment recurrent (TRec) and Treatment non-responders (TnR). Between the four treatment groups, there was no significant difference in decorin concentrations. After post-hoc Tukey’s analysis (Supplementary Table A3), decorin levels in TnR group were found significantly higher compared to the controls.

  1. Figure 5. The authors hypothesize that decorin is associated with disorders such as obesity (introduction). Perhaps would be better to establish this correlation using only the DR samples or complementing the existing figure 5 with a “only DR” correlation (which looks even milder than when combined with control).

Figures have been included in the supplementary materials (Supp Fig B1 and B2) and described in the text (Appendix A).

  1. Figure 5: Is there a sex-driven correlation of decorin with visual acuity? Is visual acuity itself having a sex-driven pattern?

Supplementary figure B2 shows that the positive correlation between decorin concentrations and visual acuity are mainly driven by the male population. This is in line with supplementary table A10 which shows that decorin concentrations in male DR subjects are significantly higher than controls. However, it is difficult to conclude if decorin correlation with visual acuity is sex-driven due to the small sample size for the females population compared to males.

  1. There exist reports of decorin levels reduced in aging and independently of pathology in many cases. Given that diabetes, and more specifically DR is in many cases age-related (T2D), could the authors explain or hypothesis why there is an increase beyond the compensatory role for angiogenesis? The global analysis of DR including non-angiogenic or low- inflammatory patients were also reported with higher decorin levels.

We agree with the reviewer. One such study where decorin concentrations in the vitreous have been reported to decline with age irrespective of the presence of glaucoma or ocular hypertension. The authors hypothesize a link between increasing active TGF-b2 and the decrease in decorin concentrations [1]. However, diabetes is a multifactorial disease that influences multiple biochemical pathways. This includes various growth factors, transcription factors, cytokines, chemokines and adhesion or matrix proteins. Future prospective studies are needed to unveil several of these non-agniogenic or low-inflammatory patients that show any corelation with decorin concentrations in the eye.

  1. There are some fundamental aspects of molecular biology needed in the context of disease and of this paper. Which cells produce decorin in the eye? And more specifically in the retina?

Decorin’s production and function in the retina have yet to be characterised.

  1. Are there reports evidencing decorin in the vitreous? Given that DR is a retina disease, most of the by-products of the pathology lead to be highly present in the vitreous (direct contact). This is an important question for 2 main reasons: 1) there are not many reports correlating aqueous to vitreous proteomics and 2) if decorin constitutes a potential biomarker, access to aqueous humor is much more feasible than vitreous, and knowing if it is correlated would accelerate research and potentially diagnosis.

Decorin concentrations have been reported to be elevated in vitreous of patients with rhegmatogenous retinal detachment (RRD) who subsequently developed proliferative vitreoretinopathy (PVR) compared to those who did not [2]. Patients with PVR associated trauma have also been reported to exhibit a significantly higher concentration of vitreous decorin 10-days post trauma [3]. It is acknowledged that vitreous concentrations will more closely reflect retinal changes due to its proximity to the retina, however, collection of vitreous is highly contraindicated due to its invasiveness, and collection is only warranted in patients undergoing vitrectomies. As such, collection and sampling of aqueous humor will be a more useful and feasible method for analysis of retinal biochemical changes. Additionally, aqueous and vitreous humor biomarkers have been reported to be directly correlated with each other [4] and changes in concentration of biomarkers in aqueous humor and tears have been used in analysis for DR severity [5,6]/ We have added this to the discussion in the text.

  1. Avoid repetition of results (stats) in discussion if already described in results.

Repetitive statistics have been removed from the discussion.

  1. Decorin in eye disease is a very interesting paradigm. Authors report very well that decorin has been described to exert anti-angiogenic abilities. Following that line, it would be logical then to find decorin more present in NPDR or in patients that respond to therapy (thus having lesser inflammatory or angiogenic profile). Why is it in then that non-responders or severe PDR cases show higher levels? Despite the authors discuss this could be a compensatory mechanism, there are not hints or evidences for this counterintuitive explanation. Are there other works showing this experimentally? Please discuss more.

Since decorin acts as an antagonist for multiple tyrosine kinase receptors and its ligands, we hypothesize that decorin is produced as a protective molecule in response to the elevated concentrations of angiogenic and inflammatory markers to curb pathological neovascularization and counterproductive inflammatory responses. We expect that following this hypothesis, elevated concentrations of angiogenic and inflammatory factors observed in the more severe cases of DR would elicit a greater decorin response compared to those observed in the less severe forms of DR. This increase in decorin concentration has been observed in other studies describing retinal damage and PVR. In kanic acid induced damage to rat retina, decorin concentrations were reported to be elevated, with strong immunoreactivities to the damaged retinal layers [7]. Additionally, PVR patients with retinal trauma have also been reported to exhibit elevated concentrations of serum and vitreal decorin [2,3]. We have modified the discussion in the text.

  1. Authors suggest in the discussion that decorin might modulate microvascular integrity, yet there is lack on data in this. Is there the possibility to extend this study to explore the imaging data of the patients and look for vascular features in the photographies?

Indeed this is a great suggestion. However, detailed analysis of the vascular parameters will require additional time and resources. We have preliminary studies using a decorin knockout mouse model that shows abnormal retinal microvasculature with vessel tortuosity and loss of pericytes in knockout mice compared to controls [8]. Also, decorin gene therapy was able to impede corneal neovascularization in rabbits [9]. These studies, along with other in vitro studies mentioned in discussion, suggests that decorin may play a role in modulating retinal microvasculature.

References:

  1. Cogan, F. de; ONeill, J.; Blanch, R.J.; Scott, R.A.H.; Logan, A. The role of Decorin in ocular ageing and disease. Invest. Ophthalmol. Vis. Sci. 2014, 55, 400–400.
  2. Begum, G.; O’neill, J.; Chaudhary, R.; Blachford, K.; Snead, D.R.J.; Berry, M.; Scott, R.A.H.; Logan, A.; Blanch, R.J. Altered decorin biology in proliferative vitreoretinopathy: A mechanistic and cohort study. Investig. Ophthalmol. Vis. Sci. 2018, 59, 4929–4936, doi:10.1167/iovs.18-24299.
  3. Abdullatif, A.M.; Macky, T.A.; Abdullatif, M.M.; Nassar, K.; Grisanti, S.; Mortada, H.A.; Soliman, M.M. Intravitreal decorin preventing proliferative vitreoretinopathy in perforating injuries: a pilot study. Graefe’s Arch. Clin. Exp. Ophthalmol. 2018, 256, 2473–2481, doi:10.1007/s00417-018-4105-7.
  4. Midena, E.; Frizziero, L.; Midena, G.; Pilotto, E. Intraocular fluid biomarkers (liquid biopsy) in human diabetic retinopathy. Graefe’s Arch. Clin. Exp. Ophthalmol. 2021, 1, 1–12, doi:10.1007/s00417-021-05285-y.
  5. Sharma, R.K.; Rogojina, A.T.; Chalam, K. V. Multiplex immunoassay analysis of biomarkers in clinically accessible quantities of human aqueous humor. Mol. Vis. 2009, 15, 60–69.
  6. Kaštelan, S.; Orešković, I.; Bišćan, F.; Kaštelan, H.; Gverović Antunica, A. Inflammatory and angiogenic biomarkers in diabetic retinopathy. Biochem. Medica 2020, 30, 1–15, doi:10.11613/BM.2020.030502.
  7. Inatani, M.; Tanihara, H.; Honjo, M.; Hangai, M.; Kresse, H.; Honda, Y. Expression of proteoglycan decorin in neural retina. Investig. Ophthalmol. Vis. Sci. 1999, 40, 1783–1791.
  8. Lim, R.R.; Gupta, S.; Grant, D.G.; Sinha, P.R.; Mohan, R.R.; Chaurasia, S.S. Retinal Ultrastructural and Microvascular Defects in Decorin Deficient (Dcn−/−) Mice. Microsc. Microanal. 2018, 24, 1264–1265, doi:10.1017/S1431927618006803.
  9. Mohan, R.R.; Tovey, J.C.K.; Sharma, A.; Schultz, G.S.; Cowden, J.W.; Tandon, A. Targeted decorin gene therapy delivered with adeno-associated virus effectively retards corneal neovascularization in vivo. PLoS One 2011, 6, doi:10.1371/journal.pone.0026432.

Reviewer 2 Report

This is a cross-sectional study looking at a modest number of subjects with diabetes and controls looking at decorin levels in the aqueous humour.

General points

  1. There are many problems with the definitions and classifications of retinopathy, maculopathy, and treatment response.
  2. The authors confuse ‘DR group’ with People with diabetes (PWD), many of whom do not have any DR. They also confuse controls for subjects with diabetes but no DR (ETDRS 0, or R0 M0)
  3. The inferences made in the discussion and conclusion are very far beyond anything that could be concluded from this study.
  4. The only strong finding is a difference in decorin between controls and PWD, but that gets forgotten in the authors attempts to overinterpret data when divided by severity.

Specific points

  1. Line 50 “DR is the leading cause of visual impairment in working adults” – please specify where you are referring to, because it is not the case everywhere. In countries with systematic screening and treatment eg UK and Iceland, DR is no longer the leading cause of visual impairment in working adults (IRD is).
  2. Line 56 Retinal detachment can occur without vitreous haemorrhage due to fibrosis and resulting traction.
  3. Line 59-61. This is a controversial point with plenty of counter evidence that optimising DM & BP control reduces the risk of progression of DR eg UKPDS. Should present the counter opinion.
  4. Lines 65 & 66-7 are contradictory: “treatment strategies for DR focus on the later stages of the disease”& “first line of defense”. Many would regard good diabetes and HT treatment as the first line of defense.
  5. Line 71 AntiVEGF has been shown to be superior laser for vision-affecting DME, so laser no longer “gold standard”
  6. Line 79-81. Decorin is presumably a target not because it is upregulated in diabetes, but because it is not upregulated enough, as it is hypothesised to be protective.
  7. Line 110 “Inclusion criteria for controls are cataract patients who had come to Narayana Nethralaya for surgery or intravitreal injections”. Intravitreal injections is not a treatment for cataract. Were other conditions included as controls?
  8. Line 113. Only neovascular form of AMD is a fibrovascular condition.
  9. Line 119. Was diabetes required as an inclusion criteria? Other conditions can mimic DR on CF, OCT & FFA.
  10. Who were reviewing CF, OCT and FFA images to make the diagnosis of DR?
  11. Inclusion and Exclusion criteria are not consistent between controls and DR patients. Any AMD, RVO or inflammatory eye disease were exclusions for controls but only AMD, RVO or uveitis causing retinal oedema for DR patients.
  12. Section 2.3. The ETDRS classification has been incorrectly transcribed (and therefore used?). Mild NPDR is MAs only (not required to be in all 4 quads); severe NPDR is MAs and haemorrhages worse than example image 2A in all 4 quads, venous beading (not bleeding) in 2 quads, or severe IRMA in 1 quad. What is “one severe microaneurysm”?
  13. Line 142. that should OR CWS rather than AND CWS
  14. Line 146 & Figure 1. The authors are trying to merge a ETDRS classification of retinopathy with a maculopathy grade (presence/absence CSMO), by adding a Class 5, which is Class 4 + VH or CSMO. But CSMO may be present at the ETDRS grades 1-3. Then they have tried to make this correspond to the Glouc grading which separates retinopathy (R) from maculopathy (M) and has a different definition of maculopathy. In this correspondence, M1 is present in ETDRS classes 1-3. I am not sure what there is to be gained by using both grading systems. I suggest the authors use one or the other and analyse retinopathy separately from maculopathy.
  15. Line 148 & 149. Omitted “Clinically Significant” before “Macular Edema”
  16. The authors say that CSMO was defined using OCT, but used a pre-OCT (ETDRS) definition. Why not use a definition based on retinal thickness measurements rather than the ETDRS definition based on stereo images/examination. How much increase in retinal thickness on OCT was defined as “retinal thickening”?
  17. Line 155 “R1 - patients with mild NPDR” whereas in Figure 1, R1 includes mild and moderate NPDR
  18. Line 173. CFT. Do the authors mean centre point thickness or central sub-field thickness?
  19. Line 173-175. The definitions of TRes and TnR are confusing. TRes requires reduction to <320um AND reduction by an absolute number (100um) – should this be ‘or’?. TnR definition uses the 320um figure then and/or a % (10%). In order to divide patient cleanly they need to use the same % or absolute figure as a cut-off for response.
  20. Also “Subjects that failed to achieve … <10 % reduction in CFT”; I think the authors mean “Subjects that failed to achieve … 10 % reduction in CFT”
  21. Section 2.5. Although earlier it is stated that subjects were recruited from patients attending for cataract surgery, it does not state when AH samples were taken. Was it just before cataract surgery was started?
  22. Need to include a table of the numbers in each retinopathy category or add to table 1.
  23. Figure 2 and text. By ‘DR group’ do authors mean patients with diabetes (PWD) because a good proportion had no DR?
  24. Line 225. The author seem to describe ETDRS class 0 (ie PWD but no DR) as controls. This is confusing them with their actual control group.
  25. Line 225. I am not familiar with Tukey’s analysis but I am suspicious of a post-hoc analysis.
  26. Line 247. What are these control groups, plural?
    Discussion
  27. Line 271. The authors mean patients with diabetes (PWD) rather than DR patients (not all had DR)
  28. Line 272-4. The association with elevated decorin was found in PWD inc those without DR/ abnormal microvasculature. So it seems to occur prior to abnormalities in the microvasculature being visible.
  29. Line 280. These data are early data on a modest sample size (with no sample size calculation). There is no difference between grades 1-4. It is therefore incorrect to state they confirm anything. They are ‘suggestive’ at best and certainly need further investigation.
  30. Line 335. The suggestion that “decorin treatment…can be analyzed [?studied] to determine its efficacy in preventing DR” on the basis of this study is jumping far ahead of the data presented here.
  31. Line 341. Again the use of “in subjects with DR” where subjects with diabetes is meant
  32. Line 342. Authors talk about decorin compensating …for neovascular remodelling but the group with the significant difference (class 5) was different on the presence of DME rather than neovascularisation (also present in class 4)
  33. Line 344. “opens new prospects for decorin-mediated therapeutics in the treatment of DR” is an overstated claim on the basis of this study.

Author Response

Reviewer 2

We thank the reviewer for the comments and suggestions, which has improved the manuscript.

Below are the answers to reviewers comments:

General points

  1. There are many problems with the definitions and classifications of retinopathy, maculopathy, and treatment response. 2. The authors confuse ‘DR group’ with People with diabetes (PWD), many of whom do not have any DR. They also confuse controls for subjects with diabetes but no DR (ETDRS 0, or R0 M0) 3. The inferences made in the discussion and conclusion are very far beyond anything that could be concluded from this study. 4. The only strong finding is a difference in decorin between controls and PWD, but that gets forgotten in the authors attempts to overinterpret data when divided by severity.

Thanks for pointing out the confusion. The definitions of retinopathy and maculopathy in the study follows the published ETDRS and Gloucestershire classifications [1,2]. The treatment response categorisation is also as per published article [3,4]. Controls are subjects without diabetes and DR. “DR” group refers to the diabetic patients who had mild to severe DR.

Specific points

  1. Line 50 “DR is the leading cause of visual impairment in working adults” – please specify where you are referring to, because it is not the case everywhere. In countries with systematic screening and treatment eg UK and Iceland, DR is no longer the leading cause of visual impairment in working adults (IRD is).

DR prevalence has been updated with more recent reports and the sentence has been modified to make it more relevant to the local population.

  1. Line 56 Retinal detachment can occur without vitreous haemorrhage due to fibrosis and resulting traction.

Agree and modified.

  1. Line 59-61. This is a controversial point with plenty of counter evidence that optimising DM & BP control reduces the risk of progression of DR eg UKPDS. Should present the counter opinion.

Indeed a good point. Counter opinion has been included.

  1. Lines 65 & 66-7 are contradictory: “treatment strategies for DR focus on the later stages of the disease”& “first line of defense”. Many would regard good diabetes and HT treatment as the first line of defense.

Text edited for clarity on DR treatment strategies.

  1. Line 71 AntiVEGF has been shown to be superior laser for vision-affecting DME, so laser no longer “gold standard”

True, a center involving DME benefits better with Anti VEGF treatment which is already proven by DRCR net studies, especially the Protocol I of DRCR.net which confirms the superior effect of PRN Anti VEGF. A focal laser does help in cases with spongy edema. We have edited the line as suggested.

  1. Line 79-81. Decorin is presumably a target not because it is upregulated in diabetes, but because it is not upregulated enough, as it is hypothesised to be protective.

This is probably the right assumption. We have modified the line and suggested decorin as an agent instead of target.

  1. Line 110 “Inclusion criteria for controls are cataract patients who had come to Narayana Nethralaya for surgery or intravitreal injections”. Intravitreal injections is not a treatment for cataract. Were other conditions included as controls?

Thanks for pointing out. We have modified the line. Controls are the patients who underwent cataract surgery.

  1. Line 113. Only neovascular form of AMD is a fibrovascular condition.

Agree and line modified as suggested.

  1. Line 119. Was diabetes required as an inclusion criteria? Other conditions can mimic DR on CF, OCT & FFA.

Yes, DM is required as an inclusion criteria. All DR subjects are patients with type 2 diabetes. Text has been modified.

  1. Who were reviewing CF, OCT and FFA images to make the diagnosis of DR?

The ophthalmologists Thirumalesh B. Mochi (TBM) and Santosh GK Gadde (SGKG), who are retina specialists, have reviewed the images.

  1. Inclusion and Exclusion criteria are not consistent between controls and DR patients. Any AMD, RVO or inflammatory eye disease were exclusions for controls but only AMD, RVO or uveitis causing retinal oedema for DR patients.

The exclusion criteria for both controls and DR patients remain the same as the retinal edema due to other predisposing conditions excluded from the study. Modified the line in the text.

  1. Section 2.3. The ETDRS classification has been incorrectly transcribed (and therefore used?). Mild NPDR is MAs only (not required to be in all 4 quads); severe NPDR is MAs and haemorrhages worse than example image 2A in all 4 quads, venous beading (not bleeding) in 2 quads, or severe IRMA in 1 quad. What is “one severe microaneurysm”?

We agree. We have modified the sentence based on suggestion.

  1. Line 142. that should OR CWS rather than AND CWS

Changes have been made to replace “and” with “or” as suggested.

  1. Line 146 & Figure 1. The authors are trying to merge a ETDRS classification of retinopathy with a maculopathy grade (presence/absence CSMO), by adding a Class 5, which is Class 4 + VH or CSMO. But CSMO may be present at the ETDRS grades 1-3. Then they have tried to make this correspond to the Glouc grading which separates retinopathy (R) from maculopathy (M) and has a different definition of maculopathy. In this correspondence, M1 is present in ETDRS classes 1-3. I am not sure what there is to be gained by using both grading systems. I suggest the authors use one or the other and analyse retinopathy separately from maculopathy.

Both the classification systems are used for the ease of description and clinical diagnosis for further analysis. We agree that Maculopathy can be present across all grades of retinopathy, which has been described in the legend as well as in the text. Therefore, having Gloucestershire classification alongside ETDRS classification helps in patient stratification.

  1. Line 148 & 149. Omitted “Clinically Significant” before “Macular Edema”

 “Clinically significant” has been removed.

  1. The authors say that CSMO was defined using OCT, but used a pre-OCT (ETDRS) definition. Why not use a definition based on retinal thickness measurements rather than the ETDRS definition based on stereo images/examination. How much increase in retinal thickness on OCT was defined as “retinal thickening”?

CSMO is a clinical diagnosis as evidenced by thickening of retina closer to macula. Retinal thickness of more than 300 microns on OCT was defined as retina thickening.

  1. Line 155 “R1 - patients with mild NPDR” whereas in Figure 1, R1 includes mild and moderate NPDR

R1 according to Gloucestershire includes both mild and moderate NPDR. We have modified the line in the text.

  1. Line 173. CFT. Do the authors mean centre point thickness or central sub-field thickness?

Yes, it means central sub-field thickness.

  1. Line 173-175. The definitions of TRes and TnR are confusing. TRes requires reduction to <320um AND reduction by an absolute number (100um) – should this be ‘or’?. TnR definition uses the 320um figure then and/or a % (10%). In order to divide patient cleanly they need to use the same % or absolute figure as a cut-off for response.

TnR definition uses the 320um figure then and/or a % (10%). In order to categorise patients, we utilized the same % or absolute figure as a cut-off for response.

  1. Also “Subjects that failed to achieve ... <10 % reduction in CFT”; I think the authors mean “Subjects that failed to achieve ... 10 % reduction in CFT”

Yes. Thanks for pointing out. Line has been edited in the text.

  1. Section 2.5. Although earlier it is stated that subjects were recruited from patients attending for cataract surgery, it does not state when AH samples were taken. Was it just before cataract surgery was started?

Yes, the AH samples were taken before the start of cataract surgery. Included in the text.

  1. Need to include a table of the numbers in each retinopathy category or add to table 1.

Sample numbers have been included in the figure 1 and 3 legends.

  1. Figure 2 and text. By ‘DR group’ do authors mean patients with diabetes (PWD) because a good proportion had no DR?

All subjects in the DR group have Type 2 diabetes with DR.

  1. Line 225. The author seem to describe ETDRS class 0 (ie PWD but no DR) as controls. This is confusing them with their actual control group.

ETDRS Class 0 = Control (no diabetes or DR)

Gloucestershire Class R0M0 = Control (no diabetes or DR)

ETDRS classes 1-5 =  mild to severe DR

Gloucestershire classes R1M0-R3M0 = mild to severe DR.

Gloucestershire classes M1 = DR with maculopathy.

  1. Line 225. I am not familiar with Tukey’s analysis but I am suspicious of a post-hoc analysis.

We have performed a post-hoc analysis to compare between each individual group after subjects have been split into their different DR classifications. Post-hoc analysis data was performed by the statisticians (Huaying Dong and Dr. Aniko Szabo) from the Department of Biostatistics, Medical College of Wisconsin, Milwaukee, WI, USA as described in the text. Post-hoc analysis can be found in supplementary tables A1, A2, A3, A6 and A9. These data allow us to further understand the relationship between the different individual groups.

  1. Line 247. What are these control groups, plural?

Discussion

We have modified the line 247.

  1. Line 271. The authors mean patients with diabetes (PWD) rather than DR patients (not all had DR)

Sorry for the confusion. We mean the DR group compared to controls. Control subjects do not have diabetes or DR. DR subjects have diabetes and DR.

  1. Line 272-4. The association with elevated decorin was found in PWD inc those without DR/ abnormal microvasculature. So it seems to occur prior to abnormalities in the microvasculature being visible.

All DR subjects had mild to severe DR. They would have abnormalities observed in mild NPDR or worse. These DR subjects have elevated decorin concentrations compared to controls (no diabetes, no DR).

  1. Line 280. These data are early data on a modest sample size (with no sample size calculation). There is no difference between grades 1-4. It is therefore incorrect to state they confirm anything. They are ‘suggestive’ at best and certainly need further investigation.

Agree. The word “confirm” has been changed to “suggest” in the text.

  1. Line 335. The suggestion that “decorin treatment...can be analyzed [?studied] to determine its efficacy in preventing DR” on the basis of this study is jumping far ahead of the data presented here.

Agree. The word “analyzed” has been changed to “studied”. We have suggested to study this in animal models.

  1. Line 341. Again the use of “in subjects with DR” where subjects with diabetes is meant

As clarified earlier, all DR subjects have mild to severe DR.

  1. Line 342. Authors talk about decorin compensating ...for neovascular remodelling but the group with the significant difference (class 5) was different on the presence of DME rather than neovascularisation (also present in class 4)

This sentence has been modified to reflect remodelling of the extracellular matrix and retinal microvasculature.

  1. Line 344. “opens new prospects for decorin-mediated therapeutics in the treatment of DR” is an overstated claim on the basis of this study.

This sentence has been replaced with future studies.

References:

  1. Stratton, I.M.; Aldington, S.J.; Taylor, D.J.; Adler, A.I.; Scanlon, P.H. A Simple Risk Stratification for Time to Development of Sight-Threatening Diabetic Retinopathy. Diabetes Care 2013, 36, 580, doi:10.2337/DC12-0625.
  2. Grading Diabetic Retinopathy from Stereoscopic Color Fundus Photographs—An Extension of the Modified Airlie House Classification: ETDRS Report Number 10. Ophthalmology 1991, 98, 786–806, doi:10.1016/S0161-6420(13)38012-9.
  3. Browning, D.J.; Stewart, M.W.; Lee, C. Diabetic macular edema: Evidence-based management. Indian J. Ophthalmol. 2018, 66, 1736, doi:10.4103/IJO.IJO_1240_18.
  4. Cai, S.; Bressler, N.M. Aflibercept, bevacizumab or ranibizumab for diabetic macular oedema: Recent clinically relevant findings from DRCR.net Protocol T. Curr. Opin. Ophthalmol. 2017, 28, 636–643, doi:10.1097/ICU.0000000000000424.

Round 2

Reviewer 1 Report

Low and colleagues have addressed most of my concerns, yet failed to integrate some of the points in the revised version of the manuscript. All and all, I appreciate the nice discussion in their response and the significantly improved manuscript. I would like to congratulate the authors on the transparency of the statistical analyses in the supplementary tables, as well as for the additional analyses performed.

The authors have presented a quite counterintuitive appendix system, and this should be fully edited and reformulated. Please follow a standard system for supplementary materials where all tables are labelled as Table S1, Table S2… and figures as Figure S1, Figure S2… (Appendix A and B is confusing). Appendix explanations should be included in the legends of tables and figures, as well as in the main text of the manuscript when pertinent (I have hardly found any references to Table AX or Figures BX in the text). It is unacceptable to include supplementary material when this is not referenced in the main text. Furthermore, tables and figures should appear in order at the supplements according to its first appearance in the main text (where data should be referenced, generally accompanying a main figure or table).

For easier comprehension of my following comments, I will refer to figures and tables by their current system in this revision. Here my additional concerns:

  1. Thanks for including a more detailed information on disease background additionally to eye disease. Despite referenced in the text, I would advise that authors create a table with this purpose where N numbers are associated to different groups (DR, CTR, responsiveness to treatment, sex, age) and additional disease information. Ideally, an excel file containing detailed data of this per patient should suffice as supplementary material.
  2. I can see now that in Figure 3 legends there is information about the specific N per group. However, authors did not include this information in Figure 1 when first cited. Please include these data in the body of the figure to allow a rapid visualization (in brackets under the specific group).
  3. Thanks for specifying the information regarding aqueous humor collection. I would advise to include in the section for author contribution the role of “performed the surgical collection of aqueous humor” by the mentioned co-authors of the study.
  4. Please include the N per group in table A10. I don’t see where table A10 is referenced in the main body of the text or where sex-driven differences are discussed in the manuscript. Despite not being significantly robust, it is fundamental to discuss this matter in clinical-related research (even when negative, this should be mentioned).
  5. Supplementary data on sex should directly complement Figure 2 and not only in the form of tables. Authors should display 3 panels representing not only the full cohort (A) violin plots but also by sex (B and C).
  6. In the Figure 3 legends, it is indicated significant p-values (*) yet this is not represented in the body of the figure. Please add it in the figure.
  7. Supplementary Tables A1 and A2 are not referenced in the main text and the data they accompany appears, for example, after Table A10. Please verify the order of the supplementary tables and refer them in the text when appropriate.
  8. In Figure 4 I still don’t understand why only one p-value is indicated in the legends and not reflected in the body of the figure. Furthermore, if all treatment groups are significantly different (to control I guess?), there should be 4 independent p-values (present in the figure as well).
  9. I agree on keeping Figure B2 in supplement, but Figure B1 should replace current Figure 5. Results regarding this should be also discusses pertinently.
  10. Figure B1 and B2 are quite interesting. It seems that control cases also correlate with decorin expression. Could the authors discuss this?
  11. I agree that sex-differences are a limitation given the cohort size and do not allow to draw many conclusions. Nevertheless, this is something important to address in the discussion or when describing the data. Please do that regarding the new analyses performed (e.g. sex).
  12. Thanks for the response regarding decorin levels according to ageing. This, however, should also be included in the manuscript together with its references. Given the little knowledge on decorin, this information is important and should be discussed.
  13. Thanks for the response on decorin expression. Lack of knowledge on this should be indicated in the manuscript as an important limitation. Is there information regarding other cells in or outside the retina expressing decorin beyond the Inatani et al. 1999 paper performed in rats? With the mounting number of single-cell repositories, authors should be able to find additional information with little effort or resources.
  14. Thank you for the discussion provided regarding aqueous vs. vitreous. Despite this has been included in the manuscript, authors should differentiate it in the discussion in two independent paragraphs. Firstly, previous evidences of decorin in disease retina should be discussed. In a second paragraph, counter-indications collecting vitreous should be discussed together with all other limitations (paragraphs about limitations of the study).

Author Response

Reviewer 1

Comments and Suggestions for Authors

Low and colleagues have addressed most of my concerns yet failed to integrate some of the points in the revised version of the manuscript. All and all, I appreciate the nice discussion in their response and the significantly improved manuscript. I would like to congratulate the authors on the transparency of the statistical analyses in the supplementary tables, as well as for the additional analyses performed.

The authors have presented a quite counterintuitive appendix system, and this should be fully edited and reformulated. Please follow a standard system for supplementary materials where all tables are labelled as Table S1, Table S2… and figures as Figure S1, Figure S2… (Appendix A and B is confusing). Appendix explanations should be included in the legends of tables and figures, as well as in the main text of the manuscript when pertinent (I have hardly found any references to Table AX or Figures BX in the text). It is unacceptable to include supplementary material when this is not referenced in the main text. Furthermore, tables and figures should appear in order at the supplements according to its first appearance in the main text (where data should be referenced, generally accompanying a main figure or table).

Thank you again for your comments and suggestions. We have updated our appendix accordingly and have mentioned them in the main text.

For easier comprehension of my following comments, I will refer to figures and tables by their current system in this revision. Here my additional concerns:

  1. Thanks for including a more detailed information on disease background additionally to eye disease. Despite referenced in the text, I would advise that authors create a table with this purpose where N numbers are associated to different groups (DR, CTR, responsiveness to treatment, sex, age) and additional disease information. Ideally, an excel file containing detailed data of this per patient should suffice as supplementary material.

We have created a table that summarizes all sample numbers (supplementary table S1). We do have a master table with each patient and is available upon request. We mentioned this under data availability statement in the manuscript. It will be unwieldy to put this large information in the manuscript.

  1. I can see now that in Figure 3 legends there is information about the specific N per group. However, authors did not include this information in Figure 1 when first cited. Please include these data in the body of the figure to allow a rapid visualization (in brackets under the specific group).

We have included the sample numbers in each figure where applicable.

  1. Thanks for specifying the information regarding aqueous humor collection. I would advise to include in the section for author contribution the role of “performed the surgical collection of aqueous humor” by the mentioned co-authors of the study.

We have included author contributions of S.G.K.G. and T.B.M for aqueous humor collection.

  1. Please include the N per group in table A10. I don’t see where table A10 is referenced in the main body of the text or where sex-driven differences are discussed in the manuscript. Despite not being significantly robust, it is fundamental to discuss this matter in clinical-related research (even when negative, this should be mentioned).

Sample numbers have been included into the table A10 which is now supplementary table S2. Discussion and references to the supplementary table has also been included in the main text (section 3.2).

  1. Supplementary data on sex should directly complement Figure 2 and not only in the form of tables. Authors should display 3 panels representing not only the full cohort (A) violin plots but also by sex (B and C).

We have included violin plots based on your suggestion.

  1. In the Figure 3 legends, it is indicated significant p-values (*) yet this is not represented in the body of the figure. Please add it in the figure.

The significant P-values were obtained from Kruskal-Wallis ANOVA analysis to determine if there are significant differences between all the groups. To avoid confusion, we have included post-hoc Tukey analysis in the body of the figure instead with P-values, which is a pairwise comparison between the different groups. Please also note that we have recalculated our statistics as the groupings have been updated as per our second reviewer’s request.

  1. Supplementary Tables A1 and A2 are not referenced in the main text and the data they accompany appears, for example, after Table A10. Please verify the order of the supplementary tables and refer them in the text when appropriate.

Supplementary tables and figures have been rearranged to follow the flow of the main text. References to the supplementary data have also been included in the text.

  1. In Figure 4 I still don’t understand why only one p-value is indicated in the legends and not reflected in the body of the figure. Furthermore, if all treatment groups are significantly different (to control I guess?), there should be 4 independent p-values (present in the figure as well).

The P-value was from Kruskal-Wallis ANOVA analysis, which was performed to determine if there are significant differences between the different groups. However, it does not tell us how different the groups are from each other. Therefore, we performed post-hoc Tukey’s analysis to further compare between each group to the control (Supp table S5). From this, we found significant difference between controls and the TnR group. We have labelled non-significant (ns) and significant groups as suggested. Also, significance level (*) obtained from post-hoc Tukey’s analysis have been included into the graph.

  1. I agree on keeping Figure B2 in supplement, but Figure B1 should replace current Figure 5. Results regarding this should be also discusses pertinently.

Thank you for the suggestion. We have modified to represent overall, control and DR correlation plots. Results have also been discussed in the discussion section. Lines 576-590.

  1. Figure B1 and B2 are quite interesting. It seems that control cases also correlate with decorin expression. Could the authors discuss this?

Discussion for correlation between control visual acuity and decorin concentrations have been included in the main text. Lines 590-594.

  1. I agree that sex-differences are a limitation given the cohort size and do not allow to draw many conclusions. Nevertheless, this is something important to address in the discussion or when describing the data. Please do that regarding the new analyses performed (e.g. sex).

We have included a discussion for sex-differences in the main text. Lines 580-584.

  1. Thanks for the response regarding decorin levels according to ageing. This, however, should also be included in the manuscript together with its references. Given the little knowledge on decorin, this information is important and should be discussed.

Information about aging has been included into the discussion. Lines 652-657.

  1. Thanks for the response on decorin expression. Lack of knowledge on this should be indicated in the manuscript as an important limitation. Is there information regarding other cells in or outside the retina expressing decorin beyond the Inatani et al. 1999 paper performed in rats? With the mounting number of single-cell repositories, authors should be able to find additional information with little effort or resources.

Lack of information regarding decorin expression in the retina has been included into the discussion. Other relevant cells expressing decorin have been mentioned. Lines 668-672

  1. Thank you for the discussion provided regarding aqueous vs. vitreous. Despite this has been included in the manuscript, authors should differentiate it in the discussion in two independent paragraphs. Firstly, previous evidences of decorin in disease retina should be discussed. In a second paragraph, counter-indications collecting vitreous should be discussed together with all other limitations (paragraphs about limitations of the study).

Thank you for the suggestion. We have divided the discussion into two independent paragraphs accordingly. Lines 646-657 and lines 659-677.

Reviewer 2 Report

There is still confusion about the participants with diabetes but no DR.  They were excluded from the control group, but they are not explicitly excluded from the diabetes group. What happened to patients with diabetes and no DR? Were they deliberately excluded? DR subjects inclusion criteria was type 2 diabetes diagnosed on fundus imaging, which is problematic because diabetes cannot be diagnosed on fundus imaging. Diabetic retinopathy can be diagnosed on fundus imaging in someone with diabetes.

ETDRS Class 0 and Glouc R0 M0 is a grading given to people with diabetes but no DR. If the authors mean controls with no diabetes, then this group should be labelled as controls.

Point 11 and 15 have not been addressed despite the authors comment.

I stand by point 18 which the authors have not countered satisfactorily.

Point 19 was misunderstood. I meant that the authors used and correctly defined Clinically Significant Macular Edema, so when they used ‘macular oedema’ later they should be specific that it was CSME, not any ME.

Point 20. So which was used. Text still gives clinical definition of CSMO, No mention of thickening being >300microns. And was 300microns subfield measurements or point measurements.

Point 22 So this needs to be clear in text. CFT is ambiguous.

Author Response

Reviewer 2

There is still confusion about the participants with diabetes but no DR.  They were excluded from the control group, but they are not explicitly excluded from the diabetes group. What happened to patients with diabetes and no DR? Were they deliberately excluded? DR subjects inclusion criteria was type 2 diabetes diagnosed on fundus imaging, which is problematic because diabetes cannot be diagnosed on fundus imaging. Diabetic retinopathy can be diagnosed on fundus imaging in someone with diabetes.

Thank you for your comments. In our year long study, we were only able to obtain four samples from patients with diabetes and no DR. We did not include them into the two arms of this study due to the small sample size. We apologize for the confusion. We have modified the text in section 2.1 as “Inclusion criteria for DR subjects included type 2 diabetes patients with clinically diagnosed DR and confirmed with fundus imaging (TRC 50DX, Topcon, Japan).”

ETDRS Class 0 and Glouc R0 M0 is a grading given to people with diabetes but no DR. If the authors mean controls with no diabetes, then this group should be labelled as controls.

Thank you for pointing this out. We have modified all groups to represent controls. This is now better explained in Tables 1 and 2 as suggested.

Point 11 and 15 have not been addressed despite the authors comment.

Point 11: We apologize for the error. The inclusion of intravitreal injections have been removed from the main text. No other conditions were included in the controls.

Point 15: We went back through the data and have found that there were no patients with other retinal conditions.

I stand by point 18 which the authors have not countered satisfactorily.

As suggested, we have removed Figure 1 and divided ETDRS classification and Gloucestershire classification into tables describing observable findings instead (Tables 1A, 1B and 2). We have also recalculated all statistics for the ETDRS classification after separating subjects based on the updated observable findings. 

Point 19 was misunderstood. I meant that the authors used and correctly defined Clinically Significant Macular Edema, so when they used ‘macular oedema’ later they should be specific that it was CSME, not any ME.

Thank you for clarifying this point. We have modified the text to represent CSME throughout the manuscript.

Point 20. So which was used. Text still gives clinical definition of CSMO, No mention of thickening being >300microns. And was 300microns subfield measurements or point measurements.

We have created new tables including ETDRS and Gloucestershire definitions of CSME (Tables 1A, 1B and 2).

Point 22 So this needs to be clear in text. CFT is ambiguous.

We have edited the main text to replace CFT with central sub-field thickness (CST).
